

# Assessing the role of anthropogenic and biogenic sources on PM₁ over Southern West Africa using aircraft measurements

Joel Brito[1], Evelyn Freney[1], Pamela Dominutti[1,2], Agnes Borbon[1], Sophie L. Haslett[3], Aurelie Colomb[1], Regis Dupuy[1], Cyrielle Denjean[4], Frederic Burnet[4], Thierry Bourriane[4], Adrien Deroubaix[5,6], Karine Sellegri[1], Hugh Coe[2], Cyrille Flamant[6], Peter Knippertz[7] and Alfons Schwarzenboeck[1]

[1]Laboratoire de Météorologie Physique, Université Clermont Auvergne, Aubière, France
[2]Instituto de Astronomia, Geofísica e Ciencias Atmosfericas, Universidade de Sao Paulo (IAG/USP), Brazil
[3]Centre of Atmospheric Sciences, School of Earth and Environmental Science, University of Manchester, Manchester, UK
[4]CNRM UMR3589 Météo France/CNRS, Toulouse France
[5]Laboratoire de Météorologie Dynamique, Ecole Polytechnique, IPSL Research University, Ecole Normale Supérieure, Université Paris-Saclay, Sorbonne Universités, UPMC Univ Paris 06, CNRS, 91128 Palaiseau, France
[6]LATMOS/IPSL, UPMC Sorbonne Universités, UPMC Univ Paris 06, CNRS and UVSQ, UMR 8190 LATMOS, Paris, France
[7]Institute of Meteorology and Climate Research, Karlsruhe Institute of Technology, Karlsruhe, Germany

*Correspondence to*: Joel Brito (j.brito@uca.fr)

**Abstract.** As part of the Dynamics-Aerosol-Chemistry-Cloud Interactions in West Africa (DACCIWA) project, an airborne campaign was designed to measure a large range of atmospheric constituents, focusing on the effect of anthropogenic emissions on regional climate. The presented study details results of the French ATR42 research aircraft, which aimed to characterize gas-phase, aerosol and cloud properties in the region during the field campaign carried out in June/July 2016 in combination with the German Falcon 20 and the British Twin Otter aircraft. The aircraft flight paths covered large areas of Benin, Togo, Ghana and Ivory Coast, focusing on emissions from large urban conurbations such as Abidjan, Accra and Lomé, as well as remote continental areas and the Gulf of Guinea. This manuscript focuses on aerosol particle measurements within the boundary layer (< 2000 m), in particular their sources and chemical composition in view of the complex mix of both biogenic and anthropogenic emissions, based on measurements from a compact time-of-flight aerosol mass spectrometer (C-ToF-AMS) and ancillary instrumentation.

Background concentrations (i.e. outside urban plumes) observed from the ATR42 indicate a fairly polluted region during the time of the campaign, with average concentrations of carbon monoxide of 131 ppb, ozone of 32 ppb, and aerosol particle number concentration (> 15nm) of 735 cm⁻³ stp. Regarding submicron aerosol composition (considering non-refractory species and Black Carbon, BC), organic aerosol (OA) is the most abundant species contributing 53 %, followed by SO₄ (27 %), NH₄ (11 %), BC (6 %), NO₃ (2%) and minor contribution of Cl (<0.5%). Average background PM₁ in the region was 5.9 µg m⁻³ stp. During measurements of urban pollution plumes, mainly focusing on the outflow of Abidjan, Accra and Lomé, pollutants are significantly enhanced (e.g. average concentration of CO of 176 ppb, and aerosol particle number concentration of 6 500 cm-3 stp), as well as PM₁ concentration (11.9 µg m⁻³ stp).



Two classes of organic aerosols were estimated based from C-ToF-AMS: particulate organic nitrates (pON) and isoprene epoxydiols secondary organic aerosols (IEPOX-SOA). Both classes are usually associated with the formation of particulate matter through complex interactions of anthropogenic and biogenic sources. During DACCIWA, pON have a fairly small contribution to OA (around 5%) and are more associated with long-range transport from central Africa than local

formation. Conversely, IEPOX-SOA provides a significant contribution to OA (around 24 % and 28 % under background and in-plume conditions). Furthermore, the fractional contribution of IEPOX-SOA is largely unaffected by changes in the aerosol composition (particularly the $SO_4$ concentration), which suggests that IEPOX-SOA concentration is mainly driven by pre-existing aerosol surface, instead of aerosol chemical properties. At times of large in-plume $SO_4$ enhancements (above 5 µg m$^{-3}$), the fractional contribution of IEPOX-SOA to OA increases above 50%, suggesting only then a change in IEPOX-SOA

controlling mechanism. It is important to note that IEPOX-SOA constitutes a lower limit to the contribution of biogenic OA, given that other processes (e.g. non-IEPOX isoprene, monoterpene SOA) are likely in the region. Given the significant contribution to aerosol concentration, it is crucial that such complex biogenic-anthropogenic interactions are taken into account in both present day and future scenario models of this fast-changing, highly sensitive region.

## 1    Introduction

Currently about 350 million people live in southern West Africa (SWA) and the population is projected to reach 800 million people by the middle of the century (Knippertz et al., 2015b), making it undoubtedly one of the least studied, most highly populated regions of the world with regard to the effects of anthropogenic activities on air quality and regional climate (Knippertz et al., 2015a). Moreover, emissions in the region from the combustion of fossil fuels, biofuels and refuse, which are already significant, are projected to rise strongly in the near future following fast-paced urbanization and population growth

(Liousse et al., 2014).

The DACCIWA (Dynamics-Aerosol-Chemistry-Cloud Interactions in West Africa) project aims to investigate the relationship between weather, climate and air pollution in SWA (Knippertz et al., 2015a). The project brings together observations from ground-based, aircraft and space-borne, as well as modelling and climate impact research. From June to July 2016 a large field campaign took place that included three inland ground supersites (Savé in Benin, Kumasi in Ghana,

Ile-Ife in Nigeria), urban sites (Cotonou in Benin, Abidjan in Ivory Coast), radiosondes and three research aircraft stationed in Lomé (Togo). Details of the field activities are given in (Flamant et al., 2017)

The atmospheric composition over SWA is known to be the result of a highly complex mix of sources. Besides the increasingly large urban emissions, the region is impacted by sea salt and oceanic compounds brought from the south by monsoon winds, Saharan dust from the north, at times large biomass burning plumes advected from the southern hemisphere,

as well as power plants, shipping emissions and oil extraction and refining platforms (Knippertz et al., 2015a; Mari et al., 2008).



In addition to the sources described above, there is about 230 000 km$^2$ of tropical forest across SWA mixed with largely deforested patches. The forest emits large quantities of Biogenic Volatile Organic Compounds (BVOCs), such as isoprene (2-methyl-1,3-butadiene, $C_5H_8$) and monoterpenes ($C_{10}H_{16}$) (Guenther et al., 2012), which can have a significant effect on atmospheric oxidative capacity (Lelieveld et al., 2008) and the formation of particulate matter (PM) (Hu et al., 2015). Ten

years prior to the DACCIWA field campaign, a large programme entitled African Monsoon Multidisciplinary Analysis (AMMA) carried out several aircraft measurements in the West African region, mainly focusing on Sahelian convection (Lebel et al., 2010) and the mesoscale convective systems of the West African Monsoon (Frey et al., 2011). Nonetheless, it allowed a significant characterization of BVOCs emission in the region (e.g. Bechara et al., 2010), including an estimate of secondary organic aerosols (SOA) formation from biogenic precursors (Capes et al., 2009). By discriminating between high and low

isoprene air masses, Capes and colleagues estimated biogenic SOA (BSOA) of remote forested areas over West Africa to be on the order of 1 µg m$^{-3}$, though the observations were close to the detection limit of the instrument. In more recent years, the identification of a range of PM formation processes from isoprene (or more generally from BVOCs) has greatly advanced (e.g. Allan et al., 2014; Liu et al., 2013; Robinson et al., 2011; Surratt et al., 2010) though there have been no recent observations over the SWA. Therefore, the large dataset gathered during the DACCIWA aircraft field campaign allows for unprecedented

characterisation of the aerosol population, comprising insights on the complex interplay between anthropogenic and biogenic sources on this highly sensitive, understudied and rapidly changing environment. In this manuscript, two important processes that lead to the formation of BSOA from BVOCs are discussed. These processes are briefly outlined here.

## 1.1    IEPOX-SOA

During daytime, isoprene reacts with hydroxyl radicals (OH) and molecular oxygen to produce isoprene hydroxyl radicals

(ISOPOO). It is currently known that these radicals can react with hydroperoxyl radicals (HO$_2$) to predominantly produce hydroxyhydroperoxides (ISOPOOH; $C_5H_{10}O_3$), or with nitric oxide (NO) to largely produce methyl vinyl ketone (MVK, $C_4H_6O$) and methacrolein (MACR, $C_4H_6O$) (Liu et al., 2013). Nitric oxide, largely emitted by urban sources, can almost entirely shift the isoprene oxidation away from the ISOPOOH pathway (Liu et al., 2016a).

The second generation products through the HO$_2$ pathway (i.e. to ISOPOOH) form isoprene epoxydiols (IEPOX) or other

hydroperoxides, which in turn can undergo reactive uptake to particles, effectively leads to particulate matter formation (Surratt et al., 2010). After reactive uptake of IEPOX, particle-phase reactions can produce several different families of species collectively labelled "IEPOX-SOA". It is important to note that the NO pathway also has a minor channel allowing the formation of IEPOX, although much less efficiently (Jacobs et al., 2014). Furthermore, non-IEPOX PM production is also possible, for example through the formation of methacrylic acid epoxide (MAE) and hydroxymethylmethyl-α-lactone

(HMML) (Kjaergaard et al., 2012; Nguyen et al., 2015) or glyoxal (Ervens and Volkamer, 2010), though in lower yields.

It is understood that the uptake of gas-phase IEPOX into the particle-phase depends on available pre-existing aerosol surface, but is also impacted by aerosol composition, which in turn affects its acidity, particle water, and nucleophilic effects. (e.g. Lin et al., 2013; Marais et al., 2016; de Sá et al., 2017; Xu et al., 2015). In regions strongly impacted by isoprene emissions,





IEPOX-SOA contributes to about a third of the observed organic aerosol (Hu et al., 2015). In order to correctly represent numerically ambient aerosol loadings, and furthermore, develop efficient abatement strategies, an understanding of the regulating mechanisms for this important class of organic aerosol is crucial.

### 1.2 Particulate Organic nitrates

The nitrate radical ($NO_3$), arising from the oxidation of nitrogen dioxide ($NO_2$) by ozone ($O_3$), is an important atmospheric radical, acting mainly at night-time due to its rapid photolysis in sunlight and its reaction with NO (Brown and Stutz, 2012). Given its formation process originating from $NO_2$ and $O_3$, the nitrate radical is directly linked to anthropogenic activities. Several BVOCs are particularly susceptible to oxidation by $NO_3$ due to one or more unsaturated functionalities, leading to the formation of organonitrates (ONs = $RONO_2$ + $RO_2NO_2$). The addition of a nitrate (-$ONO_2$) functional group to a hydrocarbon

is estimated to lower the equilibrium saturation vapour pressure by 2.5–3 orders of magnitude (Capouet and Müller, 2006), leading to potentially significant increase in particle-phase partitioning of semi-volatile species, thus contributing to PM formation (Fry et al., 2014; Nah et al., 2016; Ng et al., 2017; Perraud et al., 2012). Recent studies have identified that nighttime chemistry, particularly through the attack of BVOCs by $NO_3$ leading to formation of pON is an important source of PM over south-western United States (Xu et al., 2015) as well as over Europe (Kiendler-Scharr et al., 2016). Nonetheless, it is currently

understood that the both the formation and the lifetime of pON depends strongly on its molecular structure (Hinks et al., 2016; Nah et al., 2016), which in turn represents a highly challenging type of OA to represent numerically (Shrivastava et al., 2017). Unquestionably, observations of pON concentrations over a wide range of locations are crucial to better constrain the formation and impact of this highly uncertain aerosol type.

     The two types of SOA described above, IEPOX-SOA and pON, are typically formed through a complex series of

reactions involving both anthropogenic and biogenic compounds. The aerosol population over SWA is expected to be impacted by both types of sources, both from the mixing of emissions from its large urban conglomerates and forested areas, as well as unclear influence from other sources such as oceanic emissions, Saharan dust, advection of biomass burning and so forth. Therefore, this manuscript focuses on both quantifying submicrometric ($PM_1$) aerosol particle composition during low-level flights and, as well as assess the contribution of IEPOX-SOA and pON to the aerosol burden in the region.

## 25  2    Methods

     Measurements reported here were performed aboard the ATR-42, a French national research aircraft operated by SAFIRE (French aircraft service for environmental research). The aircraft was equipped to perform measurements of particles and gas phase species as well as cloud droplet size distribution. Aerosol particle species were sampled through a forward-facing isokinetic and isoaxial inlet with a 100% sampling efficiency for sub-micron particles and 50 % sampling efficiency for

particles with a diameter of 4.5 µm.



## 2.1 Instrumentation

The chemical composition and mass concentration of the non-refractory submicron particulate matter (NR-PM1) was measured with an Aerodyne compact time-of-flight aerosol mass spectrometer (C-ToF-AMS) with a time resolution typically of 10 s or 20 s without particle sizing information. Less often, measurements were carried out at 40 s time resolution with

sizing information (PTOF), however these will not be discussed here. Before aerosol particles were sampled by the C-ToF-AMS, they passed through a pressure controlled inlet (PCI), regulated at about 400hPa, avoiding pressure changes to the aerodynamic inlet of the instrument during airborne sampling (Bahreini et al., 2008; Freney et al., 2014). In order to extract chemically resolved mass concentrations of individual species, the C-ToF-AMS raw data were evaluated using the standard fragmentation table (Allan et al., 2004). Adjustments to the fragmentation table were made based on particle-free measurement

periods that were performed during each flight. The resolved mass concentrations included nitrate ($NO_3$), sulphate ($SO_4$), ammonium ($NH_4$), organics (Org), and chloride (Cl) species. The collection efficiency was calculated according to Middlebrook et al. (2012), usually yielding 0.5. The detection limits, considering 10s integration time, were calculated at 5 ng $m^{-3}$ for $SO_4$ and $NO_3$, 35 ng $m^{-3}$ for Cl, 45 ng $m^{-3}$ for Org and 52 ng $m^{-3}$ for $NH_4$ (Drewnick et al., 2009).

Ionization efficiency calibrations, using size-selected ammonium nitrate aerosols were carried out three times during

the field campaign. Aerosol loadings from the C-ToF-AMS were compared against volume integration using a Scanning Mobility Particle Sizer (SMPS) and Black Carbon (BC) from a Single Particle Soot Photometer (SP2), yielding good agreement (slope: 0.87, R2: 0.83, Fig. S1). The density used for each species was 1.78, 1.72, 1.72, 1.52, and 1.77 g $cm^{-3}$ for sulphate, nitrate, ammonium, chloride, and BC, respectively (Holden and Lide, 1991; Park et al., 2004). The density of organics was estimated based on the oxygen-to-carbon (O : C) and hydrogen-to-carbon (H : C) ratios (Canagaratna et al., 2015; Kuwata et

al., 2012), yielding a campaign average of 1.67 g $cm^{-3}$. Aerosol loadings from the C-ToF-AMS were also compared with measurements of C-ToF-AMS from the two other aircraft that took part in the campaign, the German DLR Falcon and the British Twin Otter (TO). For this, data were selected around take-off and landing at Lomé airport. Results between the instrument used here (ATR42) and the TO were generally in good agreement. The AMS on board of the Falcon showed considerably lower mass concentrations, an issue currently attributed to losses at the Falcon AMS pressure controlled inlet,

which is based on a different design principle than the ATR inlet.

Aerosol particle number concentration (>15 nm) were measured using an adapted TSI Condensational Particle Counter model 3010. Aerosol mass and number concentrations are corrected for standard temperature and pressure (used here 22°C and 950 hPa). Trace gases were measured by the ATR42 core chemistry instrumentation. Nitrogen oxides ($NO_x = NO+NO_2$) were measured by a TEi42 chemiluminescence detector with a blue light photolytic converter instrument (TEi42 CLT-BLC, Thermo

Fisher Scientific, Franklin, MA) with a time resolution of 1 second. The quantification of $NO_2$ is obtained directly from converting into NO using a light source emitting diodes from the blue light converter (BLC). The CLT-BLC measures $NO_2$ directly and NO indirectly after photolytic conversion by the CLT detector. The conversion efficiency adopted was 21%.



Carbon monoxide (CO) measurements were performed using the near-infrared cavity ring-down spectroscopy technique (G2401, Picarro Inc., Santa Clara, CA, USA), with a time resolution of 5 seconds.

## 2.2 Positive Matrix Factorization

Positive matrix factorization (PMF) is a statistical model that uses weighted least-square fitting for factor analysis
(Paatero and Tapper, 1994) for explaining the variability of the organic mass spectral data as linear combination of static factor profiles and their time-dependent contributions (e.g. Ulbrich et al., 2009). The PMF evaluation tool kit (PET v2.04) (Ulbrich et al., 2009) was used to prepare the data and error estimates, execute PMF and evaluate the results.

## 2.3 Particulate organic nitrate

The pON can be distinguished from inorganic nitrate by AMS technology through the fragmentation ratio of the $NO^+$
and $NO_2^+$ ions. The methodology applied here to quantify pON has been detailed elsewhere (Farmer et al., 2010; Kiendler-Scharr et al., 2016; Xu et al., 2015) and thus is presented only briefly here. The mass concentration of the nitrate functionality of organic nitrates is calculated by Eq. 2, based on the fraction of organic nitrates relative to total measured nitrate (Eq. 1), as described according to the following equations:

$$pOrgNO3_{frac} = \frac{(1 + R_{OrgNO3}) \times (R_{measured} - R_{calib})}{(1 + R_{measured}) \times (R_{OrgNO3} - R_{calib})},$$ (1)

$$pOrgNO3_{mass} = pOrgNO3_{frac} \times NO_3,$$ (2)

where $R_{measured}$ is the ratio $NO_2^+$ / $NO^+$ ions (or $m/z$ 46 and $m/z$ 30 for unit mass resolution systems), $R_{calib}$ is the ratio associated to inorganic nitrates during $NH_4NO_3$ calibrations (0.445 here) and $R_{OrgNO3}$ is 0.1. This method is considered reliable for $pOrgNO3_{frac} > 0.15$ and $pOrgNO3_{mass} > 0.1$ µg m⁻³, considering an uncertainty of 20 % (Bruns et al., 2010; Kiendler-Scharr et al., 2016) and thus such limits were considered here. Also, $pOrgNO3_{mass}$ relates to the nitrate functionality of
organic nitrates only. To account for the total particulate organic nitrate mass (here termed as pON), a molar mass of 200 g mol⁻¹ will be assumed (Kiendler-Scharr et al., 2016; Lee et al., 2016).

.

## 2.4 Isoprene Epoxydiols Secondary Organic Aerosol

Isoprene epoxydiols SOA (IEPOX-SOA) have previously been identified from AMS spectra through an enhanced
signal at ion $C_5H_6O^+$, or $m/z$ 82 in unit mass resolution systems (Allan et al., 2014; Budisulistiorini et al., 2013; Robinson et al., 2011; de Sá et al., 2017). More recently, Hu et al. (2015) proposed a diagnostic tracer for IEPOX-SOA based on datasets from a wide range of environments, such as biomass burning, urban or monoterpenes impacted areas. Results have shown that the relative contribution of $m/z$ 82 to total organic aerosol concentration, i.e., $f_{82} = (m/z \; 82 \; / \; OA)$ is a suitable tracer, with



uncertainties up to 30%. Furthermore, Hu et al. (2015) proposed an estimation of IEPOX-SOA taking in account mass concentration at $m/z$ 82 ($m_{82}$), total OA concentration ($OA$), a reference $f_{82}$ value for IEPOX-SOA ($f_{82}^{\text{IEPOX-SOA}}$ = 22‰) and a background value for $f_{82}$ ($f_{82}^{BK}$):

$$\text{IEPOX-SOA} = \frac{m_{82} - OA \times f_{82}^{BK}}{f_{82}^{\text{IEPOX-SOA}} - f_{82}^{BK}} \text{ in µg m}^{-3}, \tag{3}$$

The background values for $f_{82}$ ($f_{82}^{BK}$) has been observed by Hu et al. (2015) to range from 3‰ to 6‰ depending on the type of dominant OA source (e.g. urban, biomass burning) or air mass age, and for regions impacted by urban or biomass burning emissions is calculated as:

$$f_{82}^{BK} = 5.5 \times 10^{-3} - 8.2 \times 10^{-3} \times f_{44}, \tag{4}$$

where $f_{44} = (m/z$ 44 $/ OA)$.

### 3    Results and discussion

The DACCIWA aircraft field campaign was carried out from 27 June to 16 July 2016, during the so-called post-onset period (Phase 2 in Knippertz et al., 2017), characterized by relatively undisturbed monsoon conditions. Figure 1 depicts the ATR-42 flight trajectories below 2000 m overlaid on the forest cover in the region. Urban plumes have been mostly sampled

following north-eastward direction, following for the predominant direction of the low-level winds over the area. The spatial distribution of CO, aerosol number concentration and some aerosol species are depicted in Figure 2, showing a significant enhancement of aerosol (mass and number) downwind of the cities of Abidjan, Accra, Lomé and Cotonou. Some species (OA and CO) also show some enhancement over the Gulf of Guinea, and is most likely associated with long-range transport of biomass burning pollution from central Africa (Knippertz et al., 2017).

In order to evaluate the sources of PM$_1$ over SWA firstly a PMF analysis on the OA spectra has been carried out, and this is described in Section 3.1. A study on the impact of urban emission on aerosol composition is carried out in Section 3.2, based on a case study from flights 24/25 from 06 July 2016. Finally, Section 3.3 describes the results of a systematic identification of in-plume and regional background measurements, providing an overview of the level of different species within urban plumes and in comparison to the regional background of SWA.

### 3.1    Factor Analysis

The PMF analysis of the organic spectra was carried out using data from all ATR42 DACCIWA flights, filtered for in-cloud measurement points and limited to an altitude below 2000 m to limit the impact of free tropospheric biomass burning layers (Flamant et al., 2017), outside the scope of this work. The analysis identified three components of OA (Fig. 3), two linked with urban emissions (termed fresh and aged) and one regional component, termed Oxygenated Organic Aerosol





(OOA), following the usual nomenclature (Ulbrich et al., 2009), which is usually associated with highly aged or secondary OA (e.g. Zhang et al., 2011). It is interesting to note that Fresh Urban component of OA combines tracers of traffic emission (e.g. $m/z$ 43, 55 and 91), typical of many urban environments (e.g. Ng et al., 2010) but also include tracers of fresh biomass burning ($m/z$ 60) (Cubison et al., 2011). This result sheds light into the important role that biomass burning sources within

the city limits have on aerosol composition. This burning is thought to be mainly associated with the use of biomass as fuel, for cooking for example, or by the combustion refuse. Given the high toxicity associated with biomass burning emissions (de Oliveira Alves et al., 2015), the extensive biomass burning within city limits is likely to have an important detrimental health effect on local urban population.

Figure 4 depicts the spatial distribution of the factors, showing localised enhancements of fresh and aged urban plumes

in the outflows of the cities, as well as a more regionally homogenous distribution of the OOA factor. The latter depicts an enhancement in regions impacted by urban outflows, but also above the Gulf of Guinea, which is associated to biomass burning plumes advected from Central Africa (Flamant et al., 2017). Given the processing that has taken place during long-range transport, the mass spectrum of OA no longer carries a signature of fresh biomass burning emission (e.g. Brito et al., 2014; Cubison et al., 2011), and therefore has been grouped to the OOA component by PMF analysis, typical of aged/processed air

masses.

Despite the complex mixture of sources impacting OA concentration in the region, such as locally emitted biomass burning, biogenic and urban emissions, among others, the PMF analysis has not been able to further resolve OA sources of interest (such as IEPOX-SOA), or even to separate local fires from the urban factor. This is strongly associated with the fact that the dataset originates from airborne measurements and therefore: i) the dataset has a somewhat limited temporal coverage

(about 70h in total, compared to weeks/months/years of ground-based campaigns); ii) the dataset lacks diurnal variation, as most of the flights were either carried out during morning or afternoon hours; and iii) the aircraft samples air masses with large co-variability (e.g. biomass burning emissions from within the city itself along traffic emissions, as discussed above). The outcome of PMF results are two-fold: The different components of OA obtained from the use of PMF shall be used in a systematic identification of in-plume and background measurements; and the identification of processes of interest (IEPOX-

SOA and pON) shall be carried out using the diagnostic tracers detailed in Sections 2.3 and 2.4.

### 3.2    Case study: the Abidjan plume from 06 July 2016

On 06 July 2016, the ATR42 conducted flights in the environs of Abidjan, a city of over 4.5 million inhabitants. These flights provide an interesting case study of the effects of SWA emissions on aerosol properties, including the atmospheric

concentration of IEPOX-SOA and pON. Figure 5 shows three transects of interest, upwind Abidjan (transect 1), within the Abidjan plume (transect 2) and sampling a regional continental air mass outside of large city plumes (transect 3). Table 1 compares mean concentrations (and 5-95 % confidence interval of the mean, CI) for several species of interest. Some species had concentration values significantly lower in the advecting air mass than over continental background, such as aerosol



particle number concentration, CO, OA, NO₃, NH₄ and IEPOX-SOA. In fact, upwind Abidjan IEPOX-SOA has a CI of the mean which encompasses zero, therefore is considered negligible in this transect. The significant difference between transects 1 and 3 for typical tracers of urban emissions such as aerosol number concentration and CO, and, furthermore, the lack of IEPOX-SOA in transect 1 leads to the interpretation that upwind Abidjan air masses are mostly impacted by long-range

transport and not local recirculation. This interpretation is also corroborated by aircraft wind measurements (grey arrows in Fig. 5) and back-trajectory calculations (supplemental material Fig. S2).

In contrast to the species discussed above, BC, SO₄ and pON did not show a significant enhancement over the continental region when compared to the advecting air mass, which suggests that its regional concentrations are not being largely impacted by local emission/formation. Taken together, the changes in concentration of IEPOX-SOA and pON indicate

that the former is formed locally, whether the latter is mostly advected into the region, possibly from biomass burning, an association reported previously elsewhere (e.g. Dzepina et al., 2015; Zhang et al., 2016). Furthermore, IEPOX-SOA contributes significantly to OA concentration (mean of 27 %, CI of 21 % – 33 %), whereas the contribution of pON is minor (mean of 7 %, CI of 6 % – 9 %). It is also interesting to note that the difference in OA concentration between Continental and upwind Abidjan air masses (0.95 µg m⁻³) can be almost exclusively explained by the formation of IEPOX-SOA (0.71 µg m⁻³).

Back-trajectory analysis of transect 3 indicate a steady transport from the south, over land for about 6 h prior sampling.

When analysing the air mass in the outflow of Abidjan there is, as expected, a significant concentration enhancement for several of the species discussed here. Aerosol number concentration, for example, increases by over an order of magnitude (Table 1), whereas OA and SO₄ increase by nearly 3-fold, and BC nearly two-fold. The evolution of the Abidjan plume has also been analysed according to the plume age, calculated taking into account wind speed measured from the aircraft, and

extrapolated according to the distance from the city centre. When first crossing the Abidjan plume, at estimated three hour processing time, OA and SO₄ already depict large concentration enhancements relative to upwind Abidjan (Fig. 5a). The estimate of IEPOX-SOA also depicts a strong increase, up to 4 µg m⁻³, explaining almost 60 % of OA mass at plume age of about 3.5h. Conversely, both NO₃ and pON depict a smaller contribution to aerosol concentration, with NO₃ depicting a mean concentration of 0.54 µg m⁻³ and pON of 0.33 µg m⁻³ inside the plume.

As the plume evolves, overall concentration of OA and IEPOX-SOA tends to decrease, in contrast to SO₄ which peaks at plume age of 5.5 h. To account for dilution with plume age, the Enhancement Ratio (ER) has been calculated, i.e., the variation of the species of interest normalized by the enhancement of CO above the background (Fig. 5b). The background value of CO was chosen here to be 113 ppb, the median value upwind of Abidjan. The ER$_{IEPOX-SOA}$ tends to increases with plume age, indicating a net production of organic matter through this pathway. Conversely, ER$_{OA}$ is fairly constant with plume

processing, which suggests that the increase in IEPOX-SOA is compensated by a loss process, such as evaporation of semi-volatile species, for example. The ER$_{SO4}$ follows a similar pattern as its concentration, depicting a marked peak at about 5.5 h.

Figure 5c shows some of the diagnostic tracers of OA, namely f44, f43, f60 and f82 (Cubison et al., 2011; Hu et al., 2015). Their variability mainly follows the aged signature from the arriving air mass (high f44 in transect 1), an increasing





tendency for f82 and f44 (particularly the former) with plume age, leading to observed values of transect 3. Typical oxygen-to-carbon ratios of OA ranged from 1.43 (transect 1), 0.69 (transect 2) and 1.07 (transect 3).

In the following, a systematic analysis of plume identification through the entire dataset is described, assessing changes in aerosol properties inside and outside urban plumes within SWA.

### 3.3    In-plume enhancements and regional background levels

A systematic identification of in-plume and background air masses during DACCIWA was developed. The method is based on PMF OA apportionment, aerosol particles number concentration and a measurement location. In-plume air masses criteria were: i) Aerosol particle number concentration is above the campaign-wide 75$^{th}$ percentile, namely 2 500 cm$^{-3}$; ii) The urban factors from the PMF analysis (see Section 3.1) explain more than 70 % of OA mass concentration; iii) The distance between measurement location and the emitting city was below 110 km. Criteria i) and ii) were devised based on optimizing data statistics whereas being able to unambiguously identify the urban conurbation of origin. The distance of 110 km applied in criterion iii) was defined based on the distance between Accra and Lomé, to avoid that emissions from the latter would interfere in the plume analysis of the former. Figure S3 shows the location of the measurements identified as in-plume. From the in-plume identification analysis described above, pollution outflow from three cities were clearly identified, Lomé, Abidjan and Accra, representing 50%, 35% and 15% of the in-plume dataset.

The identification of continental background air masses were performed by filtering aerosol number concentration below the 50$^{th}$ percentile and selecting urban factors explaining less than 70% of OA. Sensitivity studies have identified that lowering these limits tended to reduce data statistics without significantly altering median values. The selection of aerosol number concentration below 50$^{th}$ or 30$^{th}$ percentile led to a decrease of background data points from 623 to 267, whereas median CO concentration would remain unchanged at 129 ppb. Similarly as the data described in the previous sections, data points used in the in-plume and regional background identification are limited to altitudes below 2 000 m.

Figure 6 and Table 2 show the mean and CI of concentrations of a range of species considered here. Regional background concentrations were elevated, with average concentration of carbon monoxide of 131 ppb, ozone of 32 ppb and aerosol particle number concentration of 735 cm$^{-3}$. Regarding PM$_1$ composition, OA was the most abundant species, contributing 54 %, followed by SO$_4$ (24 %), NH$_4$ (11 %), BC (6 %), NO$_3$ (4%) and minor contribution of Cl (<0.5%). Average background PM$_1$ in the region was 6.0 µg m$^{-3}$.

During in-plume measurements there was a marked enhancement of pollutant levels, e.g. aerosol concentration (6 500 cm$^{-3}$), CO (176 ppb), NOx (2.72 ppb) as well as PM$_1$ concentration (12.0 µg m$^{-3}$). Despite the significant enhancement in the species observed here, PM$_1$ aerosol composition is strikingly similar between background and in-plume measurements (i.e. OA - 56%, SO$_4$ - 23%, NH$_4$ - 11%, BC - 6%, and NO$_3$ - 4%) corroborating the major role that these urban conglomerates emissions have on the regional aerosol population. The analysis according to distance shown in Figure 6 depicts the clear decreasing trend for some species (e.g. aerosol number concentration and BC), whereas others have a less clear trend with ageing.



Generally, the levels of OA and $SO_4$ observed here for both background and in-plume are well within those of other aircraft field campaigns around the world that were classified as "polluted" (i.e. non-biomass burning), as described by Heald et al. (2011). Furthermore, measurements here can be compared to an aircraft campaign carried out more recently downwind of Paris (Freney et al., 2014). Although background levels over SWA are somewhat comparable with outside plume measurements in the environs of Paris (OA: 3.06 µg m$^{-3}$ over SWA and 2.2 µg m$^{-3}$ around Paris; $SO_4$: 1.67 µg m$^{-3}$ over SWA and 1.19 µg m$^{-3}$ around Paris), it is clear that in-plume concentrations increase relative to outside are more important within SWA (114% and 71% for OA and $SO_4$, respectively) than around Paris (36% and 2% for OA and $SO_4$, respectively). In the Rome metropolitan area, slightly higher levels were observed (4.5 µg m$^{-3}$ and 1.6 µg m$^{-3}$ for OA and $SO_4$, respectively) albeit with strong enhancements in these species when Saharan dust was present (Struckmeier et al., 2016).

As for IEPOX-SOA and pON, although their concentration is also enhanced within the urban plumes relative to background levels, on average their relative contribution to OA remains fairly constant (IEPOX-SOA / OA is 0.32 and 0.28 for background and in-plume, respectively, whereas pON / OA is 0.06 for both cases). It is important to note that the contribution of the IEPOX-SOA to OA represents a lower limit to biogenic OA. Other processes, such as non-IEPOX isoprene SOA or monoterpene SOA cannot be quantified under ambient measurements due to the lack of diagnostic tracers with the AMS technology. The next Section presents an analysis of the variability of IEPOX-SOA in regard to other species in the SWA.

### 3.4    IEPOX-SOA over SWA

As discussed in Section 1.1, IEPOX-SOA concentration tends to increase with $SO_4$ and decrease with NO, as observed in a number of laboratory studies (Kuwata et al., 2015; Liu et al., 2016b, 2013; Riva et al., 2016; Surratt et al., 2010). As both species, $SO_4$ and NO, originate from urban sources, the forming potential of IEPOX-SOA is the result of a complex interplay which will depend on emission strengths of each species, atmospheric chemical background (including isoprene concentration) and pre-existing aerosol properties (e.g. Marais et al., 2016).

In a somewhat similar setting to that presented here, i.e. a large urban conurbation emitting pollutants over tropical forested areas, the Manaus city plume over the Amazon rainforest has been observed to cause a net reduction of IEPOX-SOA (de Sá et al., 2017). The general interpretation for the net reduction effect in the Amazon is that although $SO_4$ concentration is enhanced by Manaus emissions, its background levels from in and out of basin sources can exceed the plume enhancement itself, and thus have a stronger controlling effect over the IEPOX-SOA forming potential. Conversely, the concentration of NO is unambiguously modulated by the Manaus emission and thus the net decreasing effect over IEPOX-SOA.

Although we show in the previous sections a significant enhancement of IEPOX-SOA within urban plumes (particularly during the Abidjan flight described in Section 3.1), it is unclear how its regional concentration responds to different $SO_4$, OA and NO concentrations. To assess the general variability throughout the region, the concentration of IEPOX-SOA, its fractional contribution to OA (termed $f_{IEPOX-SOA}$) and $NO_x$ mixing ratios are analysed as a function of $SO_4$ concentrations (Fig. 7).





Although NO would be the species expected to modulate the early stages of IEPOX formation (Section 1.1), NOx has been chosen for this analysis due to its longer lifetime, and thus considered more representative of aerosol chemical history.

Interestingly, IEPOX-SOA concentrations show a significant, steady increase with $SO_4$ across the concentration range observed during DACCIWA, seemingly unaffected by the concomitant NOx variation (which increases from 0.3 ppb to up 5.3

ppb). The $f_{IEPOX-SOA}$ shows, however, a fairly small dependency on $SO_4$ concentration up to 4 µg m$^{-3}$, with mean values around 0.23 and, above this $SO_4$ level, a sharp increase to 0.56. Although a detailed analysis of the factors controlling IEPOX-SOA concentration (e.g. acidity, particle water, aerosol surface, etc.) is outside the scope of this work, the fact that $f_{IEPOX-SOA}$ is constant despite significant changes in $NO_x$ over a wide range of $SO_4$ concentrations (up to 4 µg m$^{-3}$), is an indication that neither $NO_x$ or $SO_4$ are alone controlling the concentration of IEPOX-SOA in the region. We speculate thus that it is mainly

driven by the amount of pre-existing aerosol surface, for example (e.g. Xu et al., 2016), instead of aerosol intrinsic chemical composition. Correspondingly, the sharp increase of $f_{IEPOX-SOA}$ on the high (>4 µg m$^{-3}$) $SO_4$ range can then be interpreted as a change of driving mechanism on IEPOX-SOA formation, with $SO_4$ taking a leading role on IEPOX-SOA formation. The overall conclusion is that under background and most of the in-plume atmospheric conditions, IEPOX-SOA contributes to about 25-30 % of OA, whereas if $SO_4$ eventually has a larger contribution to PM$_1$, so will IEPOX-SOA.

**4    Summary and conclusions**

As part of the DACCIWA project, aircraft measurements were conducted over SWA during June-July 2016 with a broad objective of assessing the role of anthropogenic emissions on regional climate. Understanding the aerosol sources in the region is the first step in both being able to represent current and future scenarios in the state-of-the-art chemistry numerical models, as well as to develop efficient abatement strategies. This study focuses on aerosol sources within the atmospheric

boundary layer (<2000 m), particularly the coupling of emissions from large urban conglomerates with local biogenic emissions. PMF analysis of OA mass spectra has identified three factors, from which two are linked to urban emissions (fresh and aged) and another, more regionally homogenous highly oxygenated (OOA factor). The latter is often important, if not dominating, with background median contribution to OA of 67 %, and in-plume of 38 %.

The analysis of a case study has allowed the direct and regional impacts of SWA emissions on the aerosol composition

of an advecting air mass from the Gulf of Guinea to be inferred. Results show a significant formation of IEPOX-SOA occurs within the Abidjan urban plume (2-4 µg m$^{-3}$), where it explains the majority of OA mass. When considering observations conducted outside of the Abidjan plume, i.e. over large, mainly forested areas (representative of so-called background continental areas), IEPOX-SOA explained about 25% of the OA mass, namely 0.7 µg m$^{-3}$. It is interesting to note that the increase in OA over the forested areas in comparison to the advecting air mass can be almost entirely explained by the

formation of IEPOX-SOA (ΔOA = 0.9 µg m$^{-3}$). A similar analysis for pON has identified no quantifiable change between incoming oceanic (upwind Abidjan) and continental air masses (0.18 µg m$^{-3}$) leading to the conclusion that this species is not locally formed, but mostly advected into the region.



A systematic analysis of in-plume and regional background air masses has been carried out using the ATR42 dataset below 2000 m. Regional background concentrations are fairly polluted with average concentration of carbon monoxide of 131 ppb, ozone of 32 ppb and aerosol number concentration of 735 cm$^{-3}$. Regarding PM$_1$ composition, OA was the most abundant species, contributing 54 %, followed by SO$_4$ (24 %) and minor contribution of other species. Mean background PM$_1$ in the region was 5.9 µg m$^{-3}$. During in-plume measurements there was a marked enhancement of pollutant levels, e.g. aerosol particles number concentration (6 500 cm$^{-3}$), CO (176 ppb) and NOx (2.72 ppb), as well as PM$_1$ concentration (12.0 µg m$^{-3}$). Aerosol chemical composition is comparable between background and in-plume, likely the result of a significant impact of anthropogenic emissions scattered through the region even under the so-called background conditions.

The concentration of IEPOX-SOA has been studied according to SO$_4$ and NO$_x$ levels, in order to assess how these species might impact IEPOX-SOA concentration. Interestingly, the fractional contribution of IEPOX-SOA to OA (f$_{IEPOX-SOA}$) is fairly constant (25-30 %) for SO$_4$ concentration from 0.5 µg m$^{-3}$ up to 4 µg m$^{-3}$ (and NO$_x$ average variability between 0.5 and 2ppb). This contribution of IEPOX-SOA to OA is considered a lower limit for biogenic OA, as other species such as non-IEPOX isoprene SOA and monoterpene SOA cannot be quantified independently by the techniques employed here. For higher concentrations of SO$_4$ (>4 µg m$^{-3}$), f$_{IEPOX-SOA}$ sharply increases up to 55 %. Put together, we interpret that for SO$_4$ concentrations below 4 µg m$^{-3}$, neither NO$_x$ nor SO$_4$ seem to be significantly affecting the concentration of IEPOX-SOA in the region, and above this threshold, SO$_4$ takes a leading role on IEPOX-SOA formation. Such PM forming mechanisms must be considered in present and future scenarios, as gains from reducing primary OA emissions (such as reduction of waste burning) without reducing SO$_4$ emissions might lead to enhanced IEPOX-SOA formation, thus cancelling out possible gain in terms of PM levels. As it stands, the results presented here from the DACCIWA aircraft campaign warrants systematic long term measurements in carefully selected areas throughout SWA to assess with high degree of certainty how changes in the anthropogenic emissions profile shall impact aerosol burden in this fast-changing, highly sensitive region.

Acknowledgement

The research leading to these results has received funding from the European Union Seventh Framework Programme (FP7/2007-2013) under grant agreement n°603502. The authors would also like to extend a special thanks to the pilots and flight crew from SAFIRE for all their enthusiasm and support during the measurement campaign aboard the ATR-42 aircraft. The authors acknowledge Anneke Batenburg, Christiane Schulz, Johannes Schneider and Stephan Borrmann for the scientific input and text revision. P. Dominutti thanks the CNPq and PVE-CAPES program for financial support during her international exchange. C. Denjean thanks the Centre National des Etudes Spatiales (CNES) for financial support.





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



## Tables

**Table 1. Mean and lower and upper 95 % confidence interval of mean for different species at the flight transects shown in Figure 5.**

| Species (unit) | Advecting air mass | Abidjan plume | Continental |
|---|---|---|---|
| CO (ppb) | 113 [112 – 114] | 150 [147 – 154] | 125 [124 – 126] |
| Aerosol concentration (cm$^{-3}$) | 575 [523 – 622] | 5 340 [5 140 – 5 550] | 1 350 [1 290 – 1 420] |
| BC (µg m$^{-3}$) | 0.37 [0.31 – 0.43] | 0.50 [0.43 – 0.56] | 0.33 [0.32 – 0.35] |
| OA (µg m$^{-3}$) | 1.96 [1.82 – 2.09] | 5.90 [5.45 – 6.35] | 2.91 [2.72 – 3.10] |
| IEPOX-SOA (µg m$^{-3}$) | 0.16 [-0.23 – 0.56] | 3.14 [2.61 – 3.67] | 0.71 [0.55 – 0.86] |
| pON (µg m$^{-3}$) | 0.19 [0.15 – 0.23] | 0.33 [0.27 – 0.39] | 0.18 [0.17 – 0.21] |
| SO$_4$ (µg m$^{-3}$) | 1.39 [1.32 – 1.47] | 6.23 [5.75 – 6.71] | 1.42 [1.36 – 1.47] |
| NO$_3$ (µg m$^{-3}$) | 0.10 [0.08 – 0.13] | 0.54 [0.45 – 0.64] | 0.17 [0.15 – 0.18] |
| NH$_4$ (µg m$^{-3}$) | 0.33 [0.24 – 0.42] | 2.50 [2.27 – 2.73] | 0.21 [0.18 – 0.24] |





**Table 2. Mean and lower and upper 95 % confidence interval of mean for different species under background and in-plume conditions for ATR42 flight trajectories below 2000 m.**

| Species (unit) | Background | In-plume |
|---|---|---|
| CO (ppb) | 131 [130 – 132] | 176 [170 – 181] |
| NOx (ppb) | 0.32 [0.28 – 0.34] | 2.72 [1.84 – 3.60] |
| O$_3$ (ppb) | 31.9 [31.7 – 32.1] | 31.2 [30.5 – 32.0] |
| | | |
| Aerosol concentration (cm$^{-3}$) | 735 [725 – 745] | 6 480 [6 025 – 6 930] |
| BC (µg m$^{-3}$) | 0.34 [0.33 – 0.35] | 0.68 [0.64 – 0.72] |
| OA (µg m$^{-3}$) | 3.06 [3.00 – 3.13] | 6.56 [6.24 – 6.88] |
| IEPOX-SOA (µg m$^{-3}$) | 0.89 [0.74 – 1.04] | 1.80 [1.66 – 1.93] |
| pON (µg m$^{-3}$) | 0.17 [0.16 – 0.18] | 0.36 [0.33 – 0.39] |
| SO$_4$ (µg m$^{-3}$) | 1.67 [1.64 – 1.70] | 2.86 [2.70 – 3.03] |
| NO$_3$ (µg m$^{-3}$) | 0.12 [0.11 – 0.12] | 0.53 [0.49 – 0.57] |
| NH$_4$ (µg m$^{-3}$) | 0.66 [0.63 – 0.68] | 1.29 [1.20 – 1.38] |



**Figures**

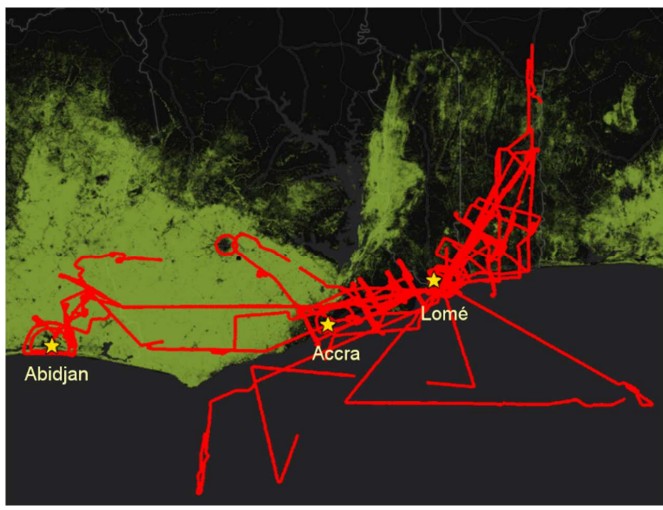

5    **Figure 1: ATR-42 trajectories (in red) during DACCIWA for altitudes below 2000m overlaid the forest cover (in green) in the region from Hansen et al., (2013).**





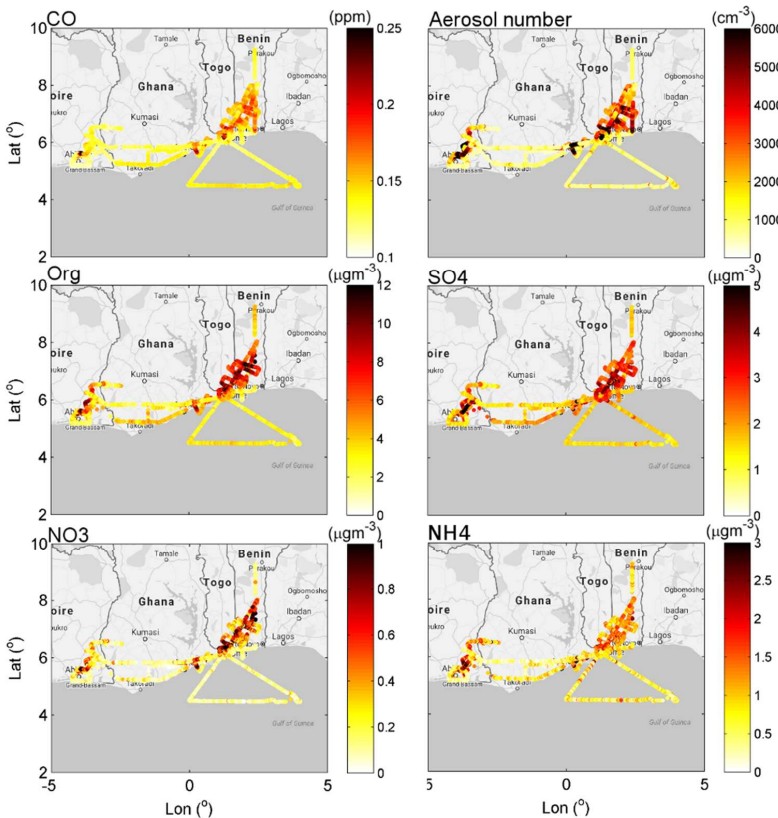

**Figure 2: Spatial distribution of CO, aerosol number concentration, OA, SO₄, NO₃ and NH₄ for flight trajectories below 2 000 m.**



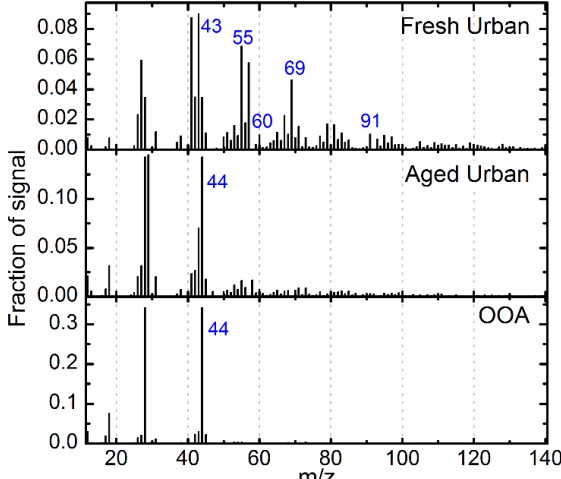

Figure 3: Mass spectra in fraction of signal of PMF factors, Fresh Urban, Aged Urban and OOA.





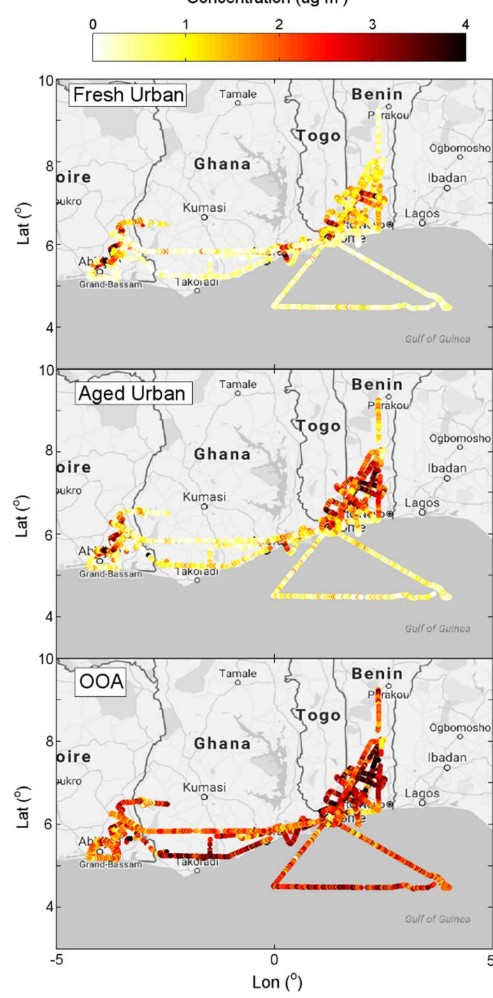

**Figure 4: Spatial distribution of OA concentration associated to three PMF factors: Fresh Urban, Aged Urban and Oxygenated Organic Aerosol.**



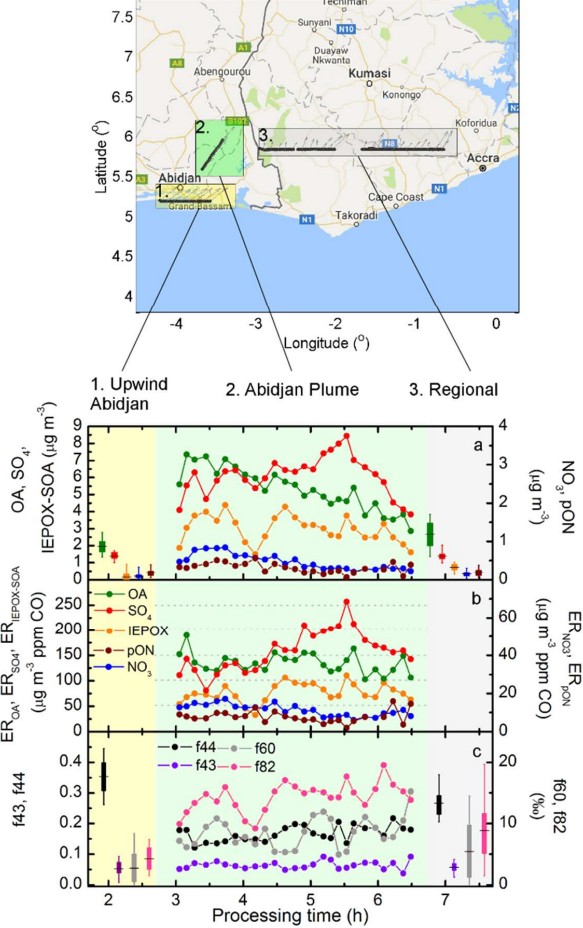

**Figure 5: Concentration (top), enhancement ratios (middle) and diagnostic ratios (bottom) for upwind of Abidjan (rectangle 1, yellow), within the plume (rectangle 2, green) and sampling regional aerosol (rectangle 3, grey). Processing time is calculated based integrated wind speed and distance from Abidjan. Grey arrows indicate wind direction measured from the aircraft. Aircraft measurements were carried out first sampling Abidjan plume around 09:30, upwind of Abidjan around 10:00, and regional aerosol at 13:30 UTC (identical to local time).**



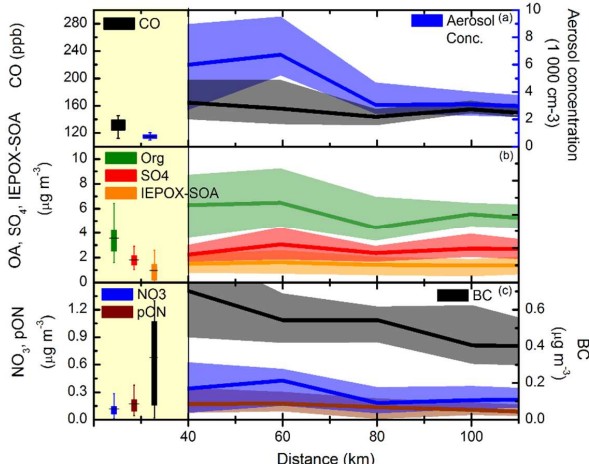

**Figure 6: Regional background (marked in yellow) and in-plume concentrations for CO and aerosol concentration (a), OA, SO₄ and IEPOX-SOA (b), NO₃, pON, and BC (c). The boxplot is the interquartile and vertical lines the 10th and 90th percentiles. The plume data show median values (line) and interquartile (shaded area) for 20 km distance bins.**



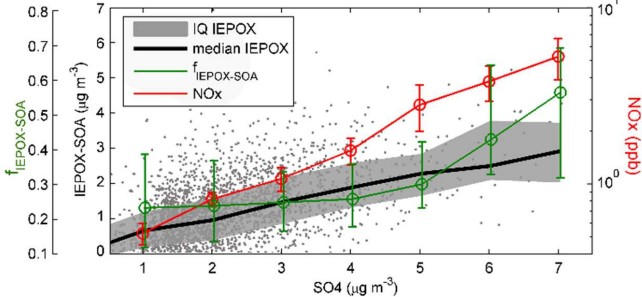

**Figure 7: Scatterplot between IEPOX-SOA concentration and SO$_4$. Black line and grey area represents mean and 5 and 95 % confidence intervals of the mean, respectively. Red and green markers are mean NO$_x$ and f$_{IEPOX-SOA}$, respectively, and range bars represent 5 and 95 % confidence intervals of the mean. The data shown here includes all ATR-42 measurements at altitudes below 2000m.**