# Peer review of "Assessing the role of anthropogenic and biogenic sources on PM1 over Southern West Africa using aircraft measurements"

_Atmospheric Chemistry and Physics, 2017_

## Referee Comment (RC1) · Anonymous Referee #1 · 20 Sep 2017

**Brito et al., 2017, ACP, Assessing the role of anthropogenic and biogenic sources on PM1 over Southern West Africa using aircraft measurements**

**General Description of manuscript:**
The authors use aerosol measurements obtained during the DACCIWA campaign to provide new insight into aerosol composition in Southwest Africa over large and rapidly growing coastal cities. The authors show that relative contributions of aerosol components (sulfate, nitrate, OA etc.) are similar for fresh and aged air, evaluate the effect of anthropogenic activity on biogenic organic aerosol, and provide suggestions for future work in a severely understudied part of the world. The content of the manuscript is appropriate for publication in ACP and should be considered after addressing the comments provided below.

**General Comments:**
This is the first time that IEPOX-SOA measurements and the relationship with anthropogenic tracers is presented for Southwest Africa. This warrants a more detailed comparison with other field experiments, beyond just comparing to Amazonia (page 11, lines 23-28). Please expand on this discussion by also evaluating the results with information from Southeast US field campaigns (for example, but not limited to, Budisulistiorini et al., 2013; 2015; Xu et al., 2015; Marais et al., 2016).

There are a number of inconsistencies that can be eliminated with a careful read-through of the text, e.g. CO is used, other times it's carbon monoxide; OA is defined, but then sometimes OA is used, other times it's organic aerosol, should it be ATR-42 or ATR42, as both are included in the text.

**Specific Comments:**
Page 1:
Line 5: Hugh Coe affiliation is not correct.

Line 29: Space between "15" and "nm"

Line 30: black carbon shouldn't be capitalized.

Line 34: Fix cm-3.

Line 25: In-text citation style for Flamant is not correct.

Page 3:
Line 3: How about including references from the AMMA study that measured and modelled these compounds in West Africa, e.g., Reeves et al. (2010), Murphy et al. (2010), Ferreira et al. (2010).

Line 4: Hu et al. (2015) is not the most appropriate reference for formation of aerosols from BVOCs. Consider instead referencing the review paper by Hallquist et al. (2009).

Lines 28-30: There is also the non-IEPOX ISOPOOH pathway that leads to SOA formation first reported in Krechmer et al. (2015).

Page 4:
Line 15: Grammar: "that the both the formation"

Page 7:
Line 1: Grammar: "taking in account" should be "taking into account" or "accounting for"

Page 8:
Line 4: "sheds light into" should be "sheds light on"

Line 7: Consider also referencing the OA health effects study by Verma et al. (2015) that showed biomass burning OA to be more toxic than other sources of OA.

Page 10:
Line 30: Better to show composition as 56% OA, 23% $SO_4$ etc.

Figure 1:
- Label countries shown, for readers not familiar with the region. Why not show the Gulf of Guinea in blue and the non-forested regions as brown or orange? Include a label for Cotonou.

Figure 5:
- The caption is misleading, as a map is presented at top. More helpful to readers to distinguish the map and the time series in the caption.
- Rectangle/partition 3 looks white, not grey.
- Add "on" in "Processing time is calculated based **on** integrated..."
- Grey arrows for wind direction are hard to see.
- Include units in the left axis of the bottom time series panel for f43 and f44.

Figure 6:
- Top right axis should be labelled "Aerosol number concentration".

Figure 7:
- In the caption the black line is the mean, but in the figure legend it is the median. Which is it? If it's the median, then why is the mean shown for the other measurements?
- What is the strength of the linear relationship between individual collocated measurements of sulfate and IEPOX SOA? How does this compare to the relationships obtained in other studies in the Southeast US and Amazonia?

**References:**
Budisulistiorini et al., Real-time Continuous Characterization of Secondary Organic Aerosol Derived from Isoprene Epoxydiols (IEPOX) in Downtown Atlanta, Georgia, using the Aerodyne Aerosol Chemical Speciation Monitor (ASCM), Environ. Sci. Technol., doi:10.1021/es400023n, 2013.
Budisulistiorini et al., Examining the Effects of Anthropogenic Emissions on Isoprene-Derived Secondary Organic Aerosol Formation During the 2013 Southern Oxidant

and Aerosol Study (SOAS) at the Look Rock, Tennessee, Ground Site, ACP, doi: :10.5194/acp-15-8871-2015, 2015.

Ferreira et al., Isoprene emissions modelling for West Africa: MEGAN model evaluation and sensitivity analysis, ACP, doi:10.5194/acp-10-8453-2010, 2010.

Hallquist et al., The formation, properties and impact of secondary organic aerosol: current and emerging issues, ACP, doi:10.5194/acp-9-5155-2009, 2009.

Krechmer et al., Formation of Low Volatility Organic Compounds and Secondary Organic Aerosol from Isoprene Hydroxyhydroperoxide Low-NO Oxidation, Environ. Sci. Technol., doi:10.1021/acs.est.5b02031, 2015.

Murphy et al., Measurements of volatile organic compounds over West Africa, ACP, doi:10.5194/acp-10-5281-2010, 2010.

Marais et al., Aqueous-phase mechanism for secondary organic aerosol formation from isoprene: application to the southeast United States and co-benefit of $SO_2$ emission controls, ACP, doi: 10.5194/acp-16-1603-2016, 2016.

Reeves et al., Chemical and aerosol characterisation of the troposphere over West Africa during the monsoon period as part of AMMA, ACP, doi: 10.5194/acp-10-7575-2010, 2010.

Verma et al., Organic Aerosols Associated with the Generation of Reactive Oxygen Species (ROS) by Water-Soluble $PM_{2.5}$, Environ. Sci. Technol., doi:10.1021/es505577w, 2015.

Xu et al., Effects of anthropogenic emissions on aerosol formation from isoprene and monoterpenes in the southeastern United States, PNAS, doi:10.1073/pnas.1417609112, 2015.

---

## Referee Comment (RC2) · Anonymous Referee #2 · 10 Oct 2017

The manuscript presented data from the DACCIWA project over south west Africa. With PMF analysis of AMS OA data, three factors were resolved: fresh and aged urban OA and a generic OOA. Using tracer methods, the contribution of IEPOX-OA and pON and the effects of anthropogenic and biogenic sources were discussed. It was found that IEPOX-OA is a major fraction of OA, contributing 24% and 29% for background and in-plume conditions, respectively.

There are very few datasets from Africa and the data presented here are interesting and will be of interest to the scientific community. The manuscript is generally wellwritten. However, I have some concerns regarding the data analysis, specifically on

the estimation of IEPOX-OA and pON concentrations.

It is difficult to understand the IEPOX-OA apportionment data. The authors noted various intrinsic reasons regarding why this factor might be challenging to resolve from PMF analysis for airborne data. However, these limitations apply to most (if not all) airborne data but other previous studies were able to resolve an IEPOX factor from flight measurements (e.g., Hu et al., ACP 2015 and references therein; Xu et al., JGR 2016). Are there something else causing a difficulty in retrieving this factor? Have the authors tested different FPEAKS in their PMF analysis, or use ME-2, or, have the authors preformed PMF analysis only on the subset of data where there might be a larger amount of IEPOX-OA (e.g., near the Abidjan area)? More analyses are needed to demonstrate and justify the results.

It is not clear why an IEPOX-OA factor cannot be resolved from PMF analysis, yet the tracer method suggests that  $\sim$ 30% (i.e., a large fraction) of the total OA is IEPOX-OA. Firstly, in PMF analysis, both variations in mass spectra and time series are taken into account. Even if the time series of IEPOX-OA are similar to other factors, the mass spectra of IEPOX-OA is very unique and has a distinctively high intensity peak at C5H6O+ (m/z 82), and this is what allows it to be resolved from other generic OOA factors in the first place. Secondly, it is possible that PMF analysis cannot resolve a factor if the contribution of that factor is too small. However, based on the tracer method, IEPOX-OA is a major fraction of ambient OA ( $\sim$ 30%). One concern is, can be concentration of IEPOX-OA be drastically overestimated in the tracer method due to the use of f82 instead of fC5H6O+ (as discussed in Hu et al., ACP 2015), given the interferences from urban and biomass burning emissions, which are also prevalent in the region? The authors should look into this further, and provide explanations and justifications regarding these drastically different results (PMF vs. tracer method).

Regarding pON analysis, it is not clear form the manuscript, but it appears that 46/30 ratios are used in the analysis instead of NO2/NO? (If not, please discard my comment below and simply clarify this in the manuscript). The approach to estimate pON re-

ACPD
quires the use of NO2/NO (or NO/NO2). The are other organic ions at m/z 30 (CH2O). Therefore, using 46/30 ratio will lead to uncertainties in the pON estimation, and yet, such uncertainty cannot be quantified as there is no way to tell the relative importance of CH2O and NO at m/z 30 for ambient data (if the instrument m/z resolution is not high enough to resolve these two ions at the same m/z). If the instrument m/z resolution is not high enough, one cannot use the R method to estimate pON with confidence.

As a large portion of the discussions and conclusions in this manuscript hinged on the IEPOX-OA (and relatively less so on pON), more analyses are needed to demonstrate the robustness of the results and conclusions in this work. Overall, I think the manuscript can be published in ACP eventually, provided that the major concerns are addressed.

Specific comments.

1. Page 3, line 11. It is noted that previous work estimated biogenic SOA of remote forested areas over west Africa is on the order of 1 ug/m3. Can the results from the current study be put in the context of this previous work? Fig. 6 appears to suggest that IEPOX-OA is about 1 ug/m3?

2. Page 4, line 14. Should be "southeastern". Also, would be appropriate to also cite Xu et al., 2015 ACP which focused on estimation of particulate organic nitrates in the southeastern US.

3. Page 14, line 17. Note that a recent review paper by Ng et al. 2017 ACP has a summary figure on "observations of pON concentrations over a wide range of locations".

4. Page 5, line 2. What is the m/z resolution of the C-ToF-AMS used in this study? Please specify clearly. Is it high enough to differentiate between the different ions at the same m/z? This has important implications for the subsequent IEPOX-OA and pON analysis.

a. Page 6, line 16. What are used in the pON analysis, NO2+ and NO+, or 46 and 30?
In the formula, R is the ratio of NO2+/NO+. If the m/z resolution of the instrument is not high enough to differentiate ions at m/z 30 (NO+ vs. CH2O+), and 46/30 ratios are used in the calculations of pON instead of NO2+/NO+ ratios, this will lead to uncertainties in the estimated pON mass concentrations. Further, as the contribution of NO+ ion to m/z 30 is unknown, one cannot tell how large the uncertainties are in the estimated pON mass concentrations. With this, if the m/z resolution of the C-ToF-AMS is not high enough, I do not think that one can use equations 1 and 2 to evaluate pON mass concentrations with confidence. Conversely, if the m/z resolution of the instrument is high enough, please simply discard the above comment specify clearly.

b. Page 6, line 23. For IEPOX-OA, the use of m/z 82 as a tracer will have a higher uncertainty than using the C56O+ ion, and can be particular sensitive to the f82 background value, which can vary widely in the presence of urban and biomass burning emissions (Hu et al., ACP 2015). The authors shall at briefly discuss the uncertainties associated with the use of f82 instead of fC5H6O+ here.

5. Page 6, line 17. Note that the RorgNO3 value = 0.1 is an assumption. This number can depend on the type of organic nitrates measured (isoprene, monoterpenes, etc) and instruments (Xu et al., ACP 2015; Kiendler-Scharr et al., GRL 2016). That the value is assumed (and not known for sure) to be 0.1 needs to be made clear in the manuscript.

6. Page 7, line 25 onwards. The authors must include some details in the SI to justify the choice of the PMF solution. How and why is a 3-factor solution chosen? Please discuss Q/Qexp, effects of seed, FPEAK, correlations of time series with external traces, correlations with reference mass spectra, etc. It is important that when one presents PMF results in a manuscript, one shall also present the details on how the specific PMF solution is chosen and clearly justify the choice of the solution.

7. Page 8, line 16 to line 25. It is noted that PMF cannot resolve an IEPOX-OA factor and the biomass burning factor. (see main comment at the beginning of review)

**ACPD**
a. Have the authors tried different FPEAK values? Or, have the authors tried using ME-2 and constrain the IEPOX-OA? Please discuss.

b. In line 26 onwards, the authors focused on IEPOX-OA in the Abidjan plume. Here, the authors noted that the IEPOX-OA accounts for 60% of total OA mass with increased plume age (line 22, page 9). With this, it is very puzzling that the tracer method results in such dominant contributions from IEPOX-OA to total OA, yet the PMF analysis cannot resolve this factor. Have the authors performed PMF analysis only on the data taken around Abidjan (Fig. 1). If IEPOX-OA indeed contributes such a large fraction of total OA near Abidjan, (and IEPOX OA has a very unique signature in AMS), I would imagine one can resolve this factor from PMF analysis of data taken around Abidjan.

8. Page 9, lines 1 and 7. CO, NH4 and BC data are discussed but not shown. Please also show the data in the figure (or in the SI if the authors deem the figure to be too busy).

9. Page 9, line 1. It is noted that NO3 concentration is also significantly lower in the advecting air mass than continental background. However, this does not seem to be case based on the data shown in Fig. 5. They are both low.

10. Page 9, line 29. It is noted that the enhancement ratio of IEPOX-OA tends to increase with plume age, indicating a net production of organic matter through this pathway. What is the mechanism for the net production with increased plume age?

11. Page 10, section 3.3. The results presented in this section are very different from the Abidjan plume.

a. Can one then assume that the large contribution of IEPOX-OA in the Abidjan plume is a special case, but not a representative of the plumes in the region? Please discuss and clarify.

b. Page 11 line 10. It is noted that IEPOX-OA and pON concentrations are also enhanced in the urban plumes. However, this statement is not consistent with the data
from Fig. 6, which seem to show that the IEPOX-OA and pON concentrations in the plume vs. background are very similar.

c. Page 11 line 10 onwards. Back on page 9 line 8, the authors noted that the changes in IEPOX-OA concentration in the advecting mass vs continental background for the Abidjan plume (i.e., no IEPOX-OA in the advecting mass) suggests that IEPOX-OA is formed locally. But in Figure 6, the concentration of IEPOX-OA in the regional background is the same as in plume, does this mean that IEPOX-OA is not formed locally?

d. Further, if I am understanding correctly, now the tracer method (f82) is applied to ALL in-plume and background data to determine IEPOX-OA concentration? Again, if the contribution is so high (page 12 line 10) at 25-30% of OA in general, it is very difficult to understand why PMF analysis did not resolve the IEPOX-OA factor.

e. Page 11, line 25. Missing de Sa et al. ACP (2017) in the reference list at the end of the manuscript.

f. Page 11, line 29. It is noted that "Although we show in the previous section a significant enhancement of IEPOX-OA within urban plumes (particularly during the Abidjan flight described in Section 3.1)". Again, data in Fig. 6 do not show a significant enhancement of IEPOX-OA in the urban plumes, and that it appears that Abidjan flight is a special case where IEPOX-OA is largely enhanced in that plume (but not for other plumes).

12. Page 12, line 3-14, discussion of Figure 7.

a. The data shown in Fig. 7 are very scattered. Nevertheless, one thing to notice is that it appears that the slope of IEPOX-OA vs. SO4 is the most similar to ground data in the SE US (Xu et al., PNAS 2015), but also falls somewhere between those observed for flight data in the SE US (Xu et al., JGR, 2016) and ground data from Amazon (de Sa et al., ACP 2017).

b. I do not understand the discussion regarding NOx. Firstly, can the authors color
the data points by NOx, similar to de Sa et al. (ACP 2017) and see if there is a trend? Secondly, it is not clear why the change in the fraction of IEPOX-OA will be interpreted as a change in the driving mechanism in IEPOX-OA formation. Based on the IEPOX-OA concentration (not fraction) vs. SO4 data, it appears that SO4 plays a role as shown in the previous studies. It is not clear why the fraction will provide specific insights regarding the formation mechanism. Please discuss and elaborate.

Technical comment. 1. Page 12, line 1, "x" in NOx should be a subscript.

---

## Author Comment (AC1) · 20 Nov 2017

We thank the anonymous reviewers for providing helpful comments and suggestions on this manuscript. The responses to the Referees comments are found below.

**Referee #1**

1. *This is the first time that IEPOX-SOA measurements and the relationship with anthropogenic tracers is presented for Southwest Africa. This warrants a more detailed comparison with other field experiments, beyond just comparing to Amazonia (page 11, lines 23-28). Please expand on this discussion by also evaluating the results with information from Southeast US field campaigns (for example, but not limited to, Budisulistiorini et al., 2013; 2015; Xu et al., 2015; Marais et al., 2016).*
   Certainly. The following paragraph has therefore been added to Page 11, L.29:

   *"The southeast US is also significantly impacted by IEPOX-SOA formation, where it explains about one-third of ambient OA in urban and rural areas (Budisulistiorini et al., 2015; Xu et al., 2015). Typically, measurements in the region have found a strong correlation of IEPOX-SOA and sulphate (e.g. Hu et al., 2015; Xu et al., 2015), and the latter has been previously hypothesized to drive IEPOX-SOA formation through nucleophilic addition leading to organosulphates (Xu et al., 2015). More recently, detailed aqueous-phase IEPOX-SOA simulation in the region has proposed that the latter is a less efficient pathway, and sulphate would be in fact enhancing IEPOX-SOA formation by increasing the aqueous aerosol volume and acidity (Marais et al., 2016). Furthermore, an important outcome has been that further reducing SO2 emissions in the region is expected to lead to a significant reduction in aerosol mass concentration via both sulphate and IEPOX-SOA (Budisulistiorini et al., 2017; Marais et al., 2016)."*

2. *There are a number of inconsistencies that can be eliminated with a careful read-through of the text, e.g. CO is used, other times it's carbon monoxide; OA is defined, but then sometimes OA is used, other times it's organic aerosol, should it be ATR-42 or ATR42, as both are included in the text.*
   Thank you for pointing it out, the text has been carefully revised and abbreviation is used after the initial definition.

3. *Page 1, Line 5: Hugh Coe affiliation is not correct.*
   The affiliation was changed accordingly.

4. *Page 1, Line 29: Space between "15" and "nm"*
   The text was changed accordingly.

5. *Page 1, Line 30: black carbon shouldn't be capitalized.*
   The text was changed accordingly.

6. *Page 1, Line 34: Fix cm-3.*
   The text was changed accordingly.

7. *Page 1, Line 25: In-text citation style for Flamant is not correct.*
   The citation was changed accordingly.

8. *Page 3, Line 3: How about including references from the AMMA study that measured and modelled these compounds in West Africa, e.g., Reeves et al. (2010), Murphy et al. (2010), Ferreira et al. (2010).*
   Please see comment #9 below.

9. *Page 3, Line 4: Hu et al. (2015) is not the most appropriate reference for formation of aerosols from BVOCs. Consider instead referencing the review paper by Hallquist et al. (2009).*
   Page 1, L1-5 has been rephrased to include the suggested references from comments #8 and #9.

   *"In addition to the sources described above, there is about 230 000 km$_2$ of tropical forest across SWA mixed with largely deforested patches. The forest in the region emits large quantities of Biogenic Volatile Organic Compounds (BVOCs), such as isoprene (2-methyl-1,3-butadiene, C5H8) (Ferreira et al., 2010; Murphy et al., 2010; Reeves et al., 2010), which can lead to a significant effect on atmospheric oxidative capacity (Lelieveld et al., 2008) and the formation of particulate matter (PM) (Claeys et al., 2004; Hallquist et al., 2009)."*

10. *Page 3, Lines 28-30: There is also the non-IEPOX ISOPOOH pathway that leads to SOA formation first reported in Krechmer et al. (2015).*
    Thank you for pointing it out, the reference was added accordingly. The updated sentence is:

    *"Furthermore, non-IEPOX PM production is also possible, for example through the formation of methacrylic acid epoxide (MAE) and hydroxymethylmethyl-α-lactone (HMML) (Kjaergaard et al., 2012; Nguyen et al., 2015), **through ISOPOOH pathway but directly forming Low Volatility Compounds (Krechmer et al., 2015)** or via glyoxal (Ervens and Volkamer, 2010), though in lower yields."*

11. *Page 4, Line 15: Grammar: "that the both the formation"*
    The text was changed accordingly.

12. *Page 7, Line 1: Grammar: "taking in account" should be "taking into account" or "accounting for"*
    The text was changed accordingly.

13. *Page 8, Line 4: "sheds light into" should be "sheds light on"*
    The text was changed accordingly.

14. *Page 8, Line 7: Consider also referencing the OA health effects study by Verma et al. (2015) that showed biomass burning OA to be more toxic than other sources of OA.*
    The text was changed accordingly.

15. *Page 10, Line 30: Better to show composition as 56% OA, 23% SO 4 etc.*
    The text was changed accordingly.

16. *Figure 1: Label countries shown, for readers not familiar with the region. Why not show the Gulf of Guinea in blue and the non-forested regions as brown or orange? Include a label for Cotonou.*

Figure was changed accordingly, countries were labelled, Cotonou was included and water surface was coloured blue. Non-forested areas were kept black though to keep a strong contrast with flight trajectories shown in red.

17. *Figure 5: The caption is misleading, as a map is presented at top. More helpful to readers to distinguish the map and the time series in the caption.*
- *Rectangle/partition 3 looks white, not grey.*
- *Add "on" in "Processing time is calculated based on integrated..."*
- *Grey arrows for wind direction are hard to see.*
- *Include units in the left axis of the bottom time series panel for f43 and f44.*

The parameters f44 and f43 do not contain units as they are shown as ratios, dimensionless, in contrast to f60 and f82 which are typically converted to ‰. Following the reviewer recommendations the figure caption was changed to the following:

*"Figure 5: Map (top) and plume analysis (bottom) for upwind of Abidjan (rectangle 1, yellow), within the plume (rectangle 2, green) and sampling regional aerosol (rectangle 3, white). Processing time is calculated based on integrated wind speed and distance from Abidjan. Aircraft measurements were carried out first sampling Abidjan plume around 09:30, upwind of Abidjan around 10:00, and regional aerosol at 13:30 UTC (identical to local time)."*

18. *Figure 6: Top right axis should be labelled "Aerosol number concentration".*

The figure was changed accordingly.

19. *Figure 7: In the caption the black line is the mean, but in the figure legend it is the median. Which is it? If it's the median, then why is the mean shown for the other measurements?*

Indeed there was an issue with the legend, as correctly pointed out by the reviewer. It has been corrected to CI and mean, as in the updated figure below:

20. *What is the strength of the linear relationship between individual collocated measurements of sulfate and IEPOX SOA? How does this compare to the relationships obtained in other studies in the Southeast US and Amazonia?*

The strength of the linear relationship between SO4 and IEPOX-SOA for our dataset (0.42) is comparable to Southeast US (0.58) and Amazonia (0.61). We have included this discussion in Page 12, L.3:

*"Interestingly, IEPOX-SOA concentrations show a significant, steady increase with $SO_4$ across the concentration range observed during DACCIWA, seemingly unaffected by the concomitant $NO_x$ variation (which increases from 0.3 ppb to up 5.3 ppb). **A linear fit between $SO_4$ and IEPOX-SOA yields a correlation coefficient of 0.42, comparable to the Southeast US (0.58, Marais et al., 2016) and Amazonia (0.61, de Sá et al., 2017), despite large differences in atmospheric background, pollution sources and sampling platform (aircraft/ground-based measurements).**"*

**Referee #2**

Major comments:

*1. The authors noted various intrinsic reasons regarding why this factor might be challenging to resolve from PMF analysis for airborne data. However, these limitations apply to most (if not all) airborne data but other previous studies were able to resolve an IEPOX factor from flight measurements (e.g., Hu et al., ACP 2015 and references therein; Xu et al., JGR 2016). Are there something else causing a difficulty in retrieving this factor? Have the authors tested different FPEAKS in their PMF analysis, or use ME-2, or, have the authors preformed PMF analysis only on the subset of data where there might be a larger amount of IEPOX-OA (e.g., near the Abidjan area)? More analyses are needed to demonstrate and justify the results. It is not clear why an IEPOX-OA factor cannot be resolved from PMF analysis, yet the tracer method suggests that ~30% (i.e., a large fraction) of the total OA is IEPOX-OA. Firstly, in PMF analysis, both variations in mass spectra and time series are taken into account. Even if the time series of IEPOX-OA are similar to other factors, the mass spectra of IEPOX-OA is very unique and has a distinctively high intensity peak at C5H6O+ (m/z 82), and this is what allows it to be resolved from other generic OOA factors in the first place. Secondly, it is possible that PMF analysis cannot resolve a factor if the contribution of that factor is too small. However, based on the tracer method, IEPOX-OA is a major fraction of ambient OA (~30%).*

Following the reviewer's suggestion, further details of the PMF analysis were included in the supplementary material, including *FPEAKS* and the number of factors. The following discussion has been included in the supplemental material:

*"Positive Matrix Factorization (PMF) has been conducted on unit mass resolution spectra of organic species for source apportionments. Organic data matrix and error matrix are generated from Squirrel software version 1.57. The PMF Evaluation Toolkit (PET) software is utilized to process the data (Ulbrich et al., 2009). Any "weak" m/z's (signal-to-noise ratio between 0.2 and 2) are downweighted by a factor of 2, and "bad" m/z's (SNR smaller than 0.2) are removed (Ulbrich et al., 2009). The PMF solutions for the dataset has been obtained following the detailed procedure described in Zhang et al., (2011).*

*For the dataset analyzed here, a 3-factor solution is chosen after carefully checking the quality of the fit parameter (Q/Qexp) (Fig. S2 and S3). Solutions with more than 3 factors depict no significant improvement resolving individual m/z's (Fig. S2), as well as display splitting behavior of existing factors instead of providing new factors (Zhang et al., 2011). The rotational ambiguity of the 3-factor solution is examined by varying the FPEAK parameter, displaying an improved correlations with external tracers and reference spectra for FPEAK=-0.4. Combining the distribution of scaled residuals for each m/z (Fig. S2), key diagnostic plots (Fig. S3), PMF solutions with characteristic mass spectral*

*signature (main text Fig. 3), and correlation with external tracers (Fig. S4), we find the 3-factor solution with FPEAK=-0.4 to be the most reasonable and meaningful solution."*

**2. *One concern is, can be concentration of IEPOX-OA be drastically overestimated in the tracer method due to the use of f82 instead of fC5H6O+ (as discussed in Hu et al., ACP 2015), given the interferences from urban and biomass burning emissions, which are also prevalent in the region? The authors should look into this further, and provide explanations and justifications regarding these drastically different results (PMF vs. tracer method).***

Actually, Hu et al., (2015) has also derived background values and analysed the validity of the methodology using the unit mass resolution signal (f82) in addition to that of C5H6O+, also under urban and biomass burning impacted areas (APPENDIX A from Hu et al, pages 11823-825). To make clearer in the manuscript that the methodology applied here (for unit mass resolution) has been carefully validated elsewhere, the sentence in P6. L26 has modified as below:

*"More recently, Hu et al. (2015) proposed a diagnostic tracer for IEPOX-SOA based on datasets from a wide range of environments, such as biomass burning, urban or monoterpenes impacted areas **for both high and unit mass resolution instruments**."*

**3. *Regarding pON analysis, it is not clear form the manuscript, but it appears that 46/30 ratios are used in the analysis instead of NO2/NO? (If not, please discard my comment below and simply clarify this in the manuscript). The approach to estimate pON requires the use of NO2/NO (or NO/NO2). The are other organic ions at m/z 30 (CH2O).Therefore, using 46/30 ratio will lead to uncertainties in the pON estimation, and yet, such uncertainty cannot be quantified as there is no way to tell the relative importance of CH2O and NO at m/z 30 for ambient data (if the instrument m/z resolution is not high enough to resolve these two ions at the same m/z). If the instrument m/z resolution is not high enough, one cannot use the R method to estimate pON with confidence.***

The authors agree that it is lacking in the manuscript the description of the instrumental mass resolution, particularly for calculation of pON, therefore the following line on P5. L1 has been modified to the following:

*"The chemical composition and mass concentration of the non-refractory submicron particulate matter (NR-PM1) was measured with an Aerodyne compact time-of-flight aerosol mass spectrometer (C-ToF-AMS), **using unit mass resolution**, and with a time resolution typically of 10 s or 20 s without particle sizing information."*

And P6. L. 16:

*"where R_measured is the ratio NO2+ / NO+ ions (or m / z 46 and m / z 30 for unit mass resolution systems, **such as used here**), R_calib is the ratio associated to inorganic nitrates during NH4NO3 calibrations (0.445 here)."*

As for the uncertainty associated to using unit mass resolution (UMR, namely mz's 30 and 46), the methodology applied here has actually been extensively used as UMR. The work mostly used as

reference for this manuscript (Kiendler-Scharr et al., 2016), has made use of several UMR instruments (9 out of 25 datasets, as described in the supplementary material, table S1). As already stated in P.6 L.18, the estimated uncertainty of the methodology by the work of Kiendler-Scharr et al., (2016), including UMR datasets, has been estimated as ±20%, the same value as referenced here.

Specific comments:

4. *Page 3, line 11. It is noted that previous work estimated biogenic SOA of remote forested areas over west Africa is on the order of 1 ug/m3. Can the results from the current study be put in the context of this previous work? Fig. 6 appears to suggest that IEPOX-OA is about 1 ug/m3?.*
Yes, the result highlighted by the reviewer indeed fits very well with our own observations. The following line has been added to Page 9, L. 14:

*"Interestingly, previous estimates of biogenic SOA over West Africa, by contrasting OA concentration in high and low isoprene air masses, has a general agreement with our results, in the order of 1 μg m-3 (Capes et al., 2009)."*

5. *Page 4, line 14. Should be "southeastern". Also, would be appropriate to also cite Xu et al., 2015 ACP which focused on estimation of particulate organic nitrates in the southeastern US..*
The text was changed accordingly.

6. *Page 5, line 2. What is the m/z resolution of the C-ToF-AMS used in this study? Please specify clearly. Is it high enough to differentiate between the different ions at the same m/z? This has important implications for the subsequent IEPOX-OA and pON analysis.*
Please refer to comments #2 and #3.

7. *Page 6, line 16. What are used in the pON analysis, NO2+ and NO+, or 46 and 30? In the formula, R is the ratio of NO2+/NO+. If the m/z resolution of the instrument is not high enough to differentiate ions at m/z 30 (NO+ vs. CH2O+), and 46/30 ratios are used in the calculations of pON instead of NO2+/NO+ ratios, this will lead to uncertainties in the estimated pON mass concentrations. Further, as the contribution of NO+ ion to m/z 30 is unknown, one cannot tell how large the uncertainties are in the estimated pON mass concentrations. With this, if the m/z resolution of the C-ToF-AMS is not high enough, I do not think that one can use equations 1 and 2 to evaluate pON mass concentrations with confidence. Conversely, if the m/z resolution of the instrument is high enough, please simply discard the above comment specify clearly.*
Please refer to comment #3.

8. *Page 6, line 23. For IEPOX-OA, the use of m/z 82 as a tracer will have a higher uncertainty than using the C56O+ ion, and can be particular sensitive to the f82 background value, which can vary widely in the presence of urban and biomass burning emissions (Hu et al., ACP 2015). The authors shall at briefly discuss the uncertainties associated with the use of f82 instead of fC5H6O+ here.*
Please refer to comment #2.

9. *Page 6, line 17. Note that the RorgNO3 value = 0.1 is an assumption. This number can depend on the type of organic nitrates measured (isoprene, monoterpenes, etc) and instruments (Xu et al., ACP 2015; Kiendler-Scharr et al., GRL 2016). That the value is assumed (and not known for sure) to be 0.1 needs to be made clear in the manuscript.*

  The sentence of P. 6, L.17 has been modified as such:

  *"The value of R_OrgNO3 has been observed to show some dependency on the molecular formula of organic nitrate, **which is unknown, and therefore was set as 0.1 similarly as Kiendler-Scharr et al. (2016)**."*

10. *Page 7, line 25 onwards. The authors must include some details in the SI to justify the choice of the PMF solution. How and why is a 3-factor solution chosen? Please discuss Q/Qexp, effects of seed, FPEAK, correlations of time series with external traces, correlations with reference mass spectra, etc. It is important that when one presents PMF results in a manuscript, one shall also present the details on how the specific PMF solution is chosen and clearly justify the choice of the solution*

  Please refer to comment #1.

11. *Page 8, line 16 to line 25. It is noted that PMF cannot resolve an IEPOX-OA factor and the biomass burning factor. (see main comment at the beginning of review). Have the authors tried different FPEAK values? Or, have the authors tried using ME-2 and constrain the IEPOX-OA? Please discuss.*

  Please refer to comment #1.

12. *In line 26 onwards, the authors focused on IEPOX-OA in the Abidjan plume. Here, the authors noted that the IEPOX-OA accounts for 60% of total OA mass with increased plume age (line 22, page 9). With this, it is very puzzling that the tracer method results in such dominant contributions from IEPOX-OA to total OA, yet the PMF analysis cannot resolve this factor. Have the authors performed PMF analysis only on the data taken around Abidjan (Fig. 1). If IEPOX-OA indeed contributes such a large fraction of total OA near Abidjan, (and IEPOX OA has a very unique signature in AMS), I would imagine one can resolve this factor from PMF analysis of data taken around Abidjan..*

  The Abidjan plume itself consists of 18 datapoints by using the 20s time resolution, upwind Abidjan amounts to 36 datapoints and regional measurements used for the case study analysis represents 138 datapoints, in total insufficient statistics to perform PMF analysis (Ulbrich et al., 2009). However, we do see a large enhancement of the unique signature (m/z 82), as shown in Figure 5 and discussed in Section 3.2. Further discussion on this topic is provided at the reply to comment #1.

13. *Page 9, lines 1 and 7. CO, NH4 and BC data are discussed but not shown. Please also show the data in the figure (or in the SI if the authors deem the figure to be too busy).*

  This data has been referred to Table 1, and the reference has been added accordingly. For completeness, however, figure S5 (shown at the end of this document) has been added to the SI depicting concentration upwind, within the Abidjan plume and regional air mass of CO, BC and NH4.

14. *Page 9, line 1. It is noted that NO3 concentration is also significantly lower in the advecting air mass than continental background. However, this does not seem to be case based on the data shown in Fig. 5. They are both low.*

Table 1 provides the lower and upper 95% confidence interval of the mean for NO3, yielding upwind Abidjan values of 0.08-0.13 µg m-3 whereas Continental air masses was 0.15-0.18 µg m-3. Although these are low concentrations in regard to other species in the region, the difference is statistically significant (given by the confidence interval). To make clearer that this sentence refers to table 1, another reference to it has been added at the end of P.9L.1.

15. *Page 9, line 29. It is noted that the enhancement ratio of IEPOX-OA tends to increase with plume age, indicating a net production of organic matter through this pathway. What is the mechanism for the net production with increased plume age?*

As stated in the main text (P.12, L.7), a detailed study of formation mechanism of IEPOX-SOA is outside the scope of this work. From a general perspective, however, chemical composition is changing with plume age, as well as aerosol volume (as clearly seen in Fig.5), and both factors are known to play an important role in IEPOX uptake (Marais et al., 2016).

16. *Page 10, section 3.3. The results presented in this section are very different from the Abidjan plume. Can one then assume that the large contribution of IEPOX-OA in the Abidjan plume is a special case, but not a representative of the plumes in the region? Please discuss and clarify.*

The goal of the study in Section 3.3 provides statistically meaningful values for the 1-month campaign (see Fig. S7 in SI). Indeed the Abidjan plume (particularly its large SO4 fraction) has not been observed elsewhere. As discussed in P.10, L.14, this dataset corresponds to Lomé, Abidjan and Accra pollution plumes.

17. *Page 11 line 10. It is noted that IEPOX-OA and pON concentrations are also enhanced in the urban plumes. However, this statement is not consistent with the data from Fig. 6, which seem to show that the IEPOX-OA and pON concentrations in the plume vs. background are very similar.*

Similarly as comment #14, it is better to refer to Table 2, where 5 and 95% confidence intervals of the mean are described, rather than visually on Fig. 6. A reference to Table 2 has been added to the text on P.11 L.10.

18. *Page 11 line 10 onwards. Back on page 9 line 8, the authors noted that the changes in IEPOX-OA concentration in the advecting mass vs continental background for the Abidjan plume (i.e., no IEPOX-OA in the advecting mass) suggests that IEPOX-OA is formed locally. But in Figure 6, the concentration of IEPOX-OA in the regional background is the same as in plume, does this mean that IEPOX-OA is not formed locally?*

Figure 6 (Section 3.3) presents only "local" measurements, namely continental background and in-plume concentrations. The case study of 3.2 has been selected because in addition to this two types of air masses, the aircraft also characterized the air mass prior arriving over continental areas of SWA, when no IEPOX-SOA has been observed, and pON presented same levels as continental air masses. This is the evidence used for stating that IEPOX-SOA is formed locally (as expected) whereas pON is mostly transported into the area.

**19.** *Further, if I am understanding correctly, now the tracer method (f82) is applied to ALL in-plume and background data to determine IEPOX-OA concentration? Again, if the contribution is so high (page 12 line 10) at 25-30% of OA in general, it is very difficult to understand why PMF analysis did not resolve the IEPOX-OA factor*

Please refer to comment #1

**20.** *Page 11, line 25. Missing de Sa et al. ACP (2017) in the reference list at the end of the manuscript.*

The reference is already listed on page 20, L. 29.

**21.** *Page 11, line 29. It is noted that "Although we show in the previous section a significant enhancement of IEPOX-OA within urban plumes (particularly during the Abidjan flight described in Section 3.1)". Again, data in Fig. 6 do not show a significant enhancement of IEPOX-OA in the urban plumes, and that it appears that Abidjan flight is a special case where IEPOX-OA is largely enhanced in that plume (but not for other plumes).*

Please refer to comment #17, there is a statistically significant enhancement of IEPOX-SOA within the plumes in comparison to continental background.

**22.** *Page 12, line 3-14, discussion of Figure 7. a. The data shown in Fig. 7 are very scattered. Nevertheless, one thing to notice is that it appears that the slope of IEPOX-OA vs. SO4 is the most similar to ground data in the SE US (Xu et al., PNAS 2015), but also falls somewhere between those observed for flight data in the SE US (Xu et al., JGR, 2016) and ground data from Amazon (de Sa et al., ACP 2017).*

Please refer to comment #20 from reviewer 1.

**23.** *I do not understand the discussion regarding NOx. Firstly, can the authors color the data points by NOx, similar to de Sa et al. (ACP 2017) and see if there is a trend? Secondly, it is not clear why the change in the fraction of IEPOX-OA will be interpreted as a change in the driving mechanism in IEPOX-OA formation. Based on the IEPOX-OA concentration (not fraction) vs. SO4 data, it appears that SO4 plays a role as shown in the previous studies. It is not clear why the fraction will provide specific insights regarding the formation mechanism. Please discuss and elaborate.*

As discussed in Section 1.1, NO leads to the suppression of IEPOX-SOA (e.g. Liu et al., 2016), and therefore it is of interest to analyse how IEPOX-SOA (and its fraction) responds to different concentrations of SO4 and NO. Further discussion is presented in P. 12, L.8.

**24.** *Technical comment. 1. Page 12, line 1, "x" in NOx should be a subscript.*

The text has been modified accordingly.

**References**

[revised manuscript text omitted]

factors. (e) Variations of the residual (= measured - reconstructed) of the least-square-fit as a function of time. (f) The Q/Qexp for each point as a function of time. (g) The Q/Qexp values for each m/z.

**Figure S4**

[Figure]

**Fig. S4.** Correlation with external tracers for the 3 factor solution. FPEAK was chosen as -0.4 to maximize correlation with external tracers and reference spectra (referring to Fig. 3 of main text).

**Figure S6**

[Figure]

**Fig. S6.** Concentration (top) and enhancement ratios (bottom) for upwind of Abidjan (yellow), within the plume (green) and sampling regional aerosol (grey). Aircraft measurements were carried out first sampling Abidjan plume around 09:30, upwind of Abidjan around 10:00, and regional aerosol at 13:30 UTC (identical to local time).